# Influences of Technological Parameters on Cross-Flow Nanofiltration of Cranberry Juice

**DOI:** 10.3390/membranes11050329

**Published:** 2021-04-29

**Authors:** Dat Quoc Lai, Nobuhiro Tagashira, Shoji Hagiwara, Mitsutoshi Nakajima, Toshinori Kimura, Hiroshi Nabetani

**Affiliations:** 1Department of Food Technology, Faculty of Chemical Engineering, Ho Chi Minh City University of Technology (HCMUT), 268 Ly Thuong Kiet Street, District 10, Ho Chi Minh City 72506, Vietnam; 2Vietnam National University Ho Chi Minh City, Linh Trung Ward, Thu Duc District, Ho Chi Minh City 71308, Vietnam; 3AOHATA Corporation, 1-1-25 Tadanouminakamachi Takehara-shi, Hiroshima 729-2392, Japan; Nobuhiro_Tagashira@aohata.co.jp; 4Food Research Institute, NARO, 2-1-12 Kannondai, Tsukuba, Ibaraki 305-8642, Japan; shoji@affrc.go.jp; 5Faculty of Life and Environmental Sciences, University of Tsukuba, Tennodai, Tsukuba, Ibaraki 305-8577, Japan; nakajima.m.fu@u.tsukuba.ac.jp; 6Research Faculty of Agriculture, Hokkaido University, Sapporo 050-8589, Japan; toshinorikmr@hotmail.com; 7National Agriculture and Food Research Organization Faculty of Home Economics, Food Research Institute, Tokyo Kasei University, 1-18-1 Kaga, Itabashi, Tokyo 173-8602, Japan; nabetani-h@tokyo-kasei.ac.jp

**Keywords:** nanofiltration, cranberry juice, benzoic acid, feed flow rate, negative rejection

## Abstract

The paper focused on the influence of operative conditions on the separation of benzoic acid from 10 °Brix cranberry juice by cross-flow nanofiltration with a plate and frame pilot scale (DDS Lab Module Type 20 system). Six kinds of commercial nanofiltration membrane were investigated. The results showed that the rejection of benzoic acid was significantly lower than that of other components in cranberry juice, including sugars and other organic acids. In a range of 2–7.5 L/min, feed flow rate slightly affected the performance of nanofiltration. Higher temperatures resulted in higher permeate flux and lower rejection of benzoic acid, whereas rejection of sugar and organic acid was stable at a high value. In a range of 2.5–5.5, pH also significantly affected the separation of benzoic acid and negative rejection against benzoic acid was observed at pH 4.5 with some of the membranes. This implies that pH 4.5 is considered as an optimum pH for benzoic acid separation from cranberry juice. The lower permeate flux caused a lower rejection of benzoic acid and negative rejection of benzoic acid was observed at the low permeate flux. Pretreatment by ultrafiltration with CR61PP membranes could improve the permeate flux but insignificantly influenced the efficiency of separation. The results also indicated that NF99 and DK membranes can be effectively used to separate benzoic acid from cranberry juice.

## 1. Introduction

Benzoic acid (M = 122.12, pKa = 4.21) has been used widely in the food and cosmetic industries as a preservative because of its anti-bacterial property. In nature, it occurs in some fruits and spices, especially in the cranberry (*Vaccinium macrocarpon*) [1]. The amount of benzoic acid in cranberry fruit is approximately 4.741 g/kg fresh weight, including about 10% in free state and 90% in bound state [2]. Fresh cranberry juice (prepared from the cranberry by squeezing) contains 41 ppm of free benzoic acid, and the total content (including free and bound states) is about 178 ppm [1,3]. The content of benzoic acid in 50 °Brix concentrated cranberry is approximately 500 ppm. This content implies that cranberry juice contains an excessive amount of benzoic acid which can be utilized as a natural preservative if it is separated with low cost.

Nanofiltration is a pressure-driven membrane process situated for retain compounds of molecular weight up to 150–250 g·mol^−1^ and charged molecules, especially multivalent ions. It has a pore size in the range of 0.2–2 nm with a molecular weight cut-off (MWCO) from 200 to 1000 Da [4,5]. In recent decades, it has been popularly applied in the food industry for fractionating sugars [6], the recovery and purification of amino acids [7], the recovery and purification of fermentation products [8], the concentration of coffee extract [9], juices [10], and the rejection of salt in whey protein production [11]. The reasons for this are the advantages of nanofiltration such as: low operating temperature, low operation cost, and being simple to install and maintain. 

Separation by nanofiltration is achieved by size and charge exclusion [12]. The transport of solutes through nanofiltration membranes due to three mechanisms: convection, diffusion, and electromigration [13]. The size exclusion is dependent on the membrane structure (i.e., pore size and porosity), with a denser structure leading to less permeation, whereas charge exclusion depends on the charge of membrane and solute, as well as the ion strength of the solution [14]. 

In nanofiltration, pH is one of the most important factors affecting separation [13]. It influences the charge of membranes, which can lead to changes in their pore side and charge exclusion of membranes because of electrostatic interaction, not only between charged groups in membranes but also between these groups and charged components in the solution. Besides, in the case of weak organic acids such as benzoic acid, pH also determines the dissociation. The dissociated solute is charged, and its separation depends on charge in the membrane. 

In addition, temperature affects the performance of separation by nanofiltration [13]. This is attributed to the effect of temperature on viscosity, the sorption of solute on membranes, and the mobility of polymer chains. Consequently, it leads to changes in the structure of membranes and the physical–chemical properties of solutions. 

A severe problem in nanofiltration is the decline in permeate flux in the operation, which is caused by concentration polarization and fouling. Concentration polarization leads to the boundary layer formation on a membrane’s surface. During nanofiltration, some compounds can be adsorbed on the membrane’s surface and pore wall, leading to fouling. This, along with the boundary layer, makes it resistant to permeate flux increase. This also has an effect on the separation. However, these phenomena can be diminished by increasing cross-flow velocity. The pretreatment by ultrafiltration also helps to limit fouling because macromolecules are rejected and the cake formation, which can cause fouling, is reduced. 

Previous work has indicated that benzoic acid can be significantly separated from cranberry juice by nanofiltration with a dead-end bench scale system [15]. In this work, the influence of technological parameters, (including the feed flow rate, temperature, pH, and operation pressure) on the separation of benzoic acid from cranberry juice by cross-flow nanofiltration is focused with a cross-flow pilot scale system. The effect of pretreatment by ultrafiltration on the performance of nanofiltration is also investigated to evaluate its application for limiting fouling. 

## 2. Materials and Methods

### 2.1. Materials

Cranberry juice was supplied by Aohata Company (Hiroshima, Japan). The 50 °Brix concentrated cranberry juice was diluted using deionized water to get 10 °Brix cranberry juice. All chemicals were supplied by Wako Pure Chemical Industries, Ltd. (Osaka, Japan) with analytical grade. pH of cranberry juice was adjusted by NaOH. 

### 2.2. Membrane Apparatus

Six kinds of commercial nanofiltration membrane were investigated with the characteristics shown in Table 1.

The De Danske Sukkerfabriker (DDS) “Lab Module Type 20” plate and frame system (Copenhagen, Denmark) was used to conduct the nanofiltration in this study (Figure 1). The unit consisted of 6 couples of membrane sheets (0.018 m^2^/sheet). The unit was equipped with a high-pressure pump (Hydra-Cell pump, supplied by Wanner Engineering Inc., Minneapolis, MN, USA) to provide feed for the unit. Six kinds of commercial nanofiltration membranes (stated in Table 1) were installed in series (1 couple per membrane). During operation, permeate and retentate were fully circulated into the feed tank to remain in the feedwithout changing its chemical composition (total recirculation mode).

### 2.3. Ultrafiltration Pretreatment

The 10 °Brix cranberry juice was ultrafiltrated by a GR61PP membrane with “Lab Module Type 20” plate and frame system (Figure 1), under operation: feed flow rate: 7.5 L/min, operating pressure: 3 bar, temperature: 25 °C. 

### 2.4. Analysis Methods

Glucose and fructose (180 g/mol) were analyzed by using YMC-Pack Polyamine II (250 × 4.6 mm ID) (supplied by YMC Co. Ltd., Kyoto, Japan) coupled to a refractive index detector and Waters 515 HPLC pump. The column was maintained at 35 °C and the mobile phase was acetonitrile/water (70/30) at the flow rate 1.0 mL/min. 

Benzoic acid (122 g/mol) was analyzed by using YMC—Pack Pro C18 (150 × 4.6 mm ID) (supplied by YMC Co. Ltd., Kyoto, Japan) and the Waters Alliance 2695 HPLC system coupled to Water 2487 UV detector (UV 210–240 nm). The column was maintained at 45 °C and the mobile phase was methanol/phosphate buffer pH 4.5 (25/75) at the flow rate of 0.9 mL/min.

Quinic acid (192 g/mol), malic acid (134 g/mol) and citric acid (192 g/mol) were analyzed by using fused silica capillary (l = 72 cm, L = 80.5, ID = 50 µm) and a capillary electrophoresis system (Agilent G1600A) coupled to the detector: Sig. = 350/20 nm, Ref. = 275/10 nm. The buffer was Agilent plating bath buffer for CE (part No. 5064–8236) and the injection was 6 s × 50 mbar. The capillary was maintained at 15 °C. The voltage was −25 kV.

### 2.5. Performance Parameters

The performance of nanofiltration was expressed in terms of permeate flux (L/m^2^/h) and the observed rejection of benzoic acid, sugars (being defined as the sum of glucose and fructose), organic acids (being defined as the sum of quinic acid, malic acid, and citric acid). Both of them were determined when permeate flux became stable (after approximately 30–40 min).

The observed rejection $R_{o}$ was calculated from the following formula
(1)$$R_{o}=1-\frac{C_{p}}{C_{f}}$$
where, $C_{p}$ and $C_{f}$ were the concentration (g/L) of the solutes in permeate and feed, respectively.

## 3. Results

### 3.1. Influence of Feed Flow Rate

The effect of feed flow rate on permeate flux in the separation of benzoic acid from cranberry juice was shown in Figure 2a. It is apparent that feed flow rate slightly affected the permeate flux. Theoretically, the increase in cross-flow velocity reduces the concentration polarization on the membrane surface. When the cross-flow velocity reaches a critical value, the concentration polarization is considered to be eliminated and the relationship between solution permeability and applied pressure is linear at constant cross-flow velocity [18]. In our present work, with the investigated range of feed flow rate, during the nanofiltration of cranberry juice with “Lab Module Type 20” plate and frame system, concentration polarization might be diminished because the relationship between permeate flux and applied pressure was linear, with the correlation coefficient being approximately 1 (the result was not shown in this paper). Therefore, the possible explanation for the effect of feed flow rate is that an increase in velocity of fluid flow on the membrane surface reduced the reversible adsorption on the membrane surface, consequently reducing the resistance to permeate flux.

From the data in Figure 2a, DRA 4510 has almost half the flux of NF99 but NF99 has less permeate flux compared to DRA 4510. This is because NF99 is a hydrophilic membrane, while DRA 4510 is a hydrophobic membrane. Hydrophilic membranes are capable of forming gravitational interactions between water and membrane materials such as dipole–dipole interactions, hydrogen bonds and ionic–dipole interactions [19].

The effect of feed flow rate on the rejection of sugars (Figure 2b) and organic acids (Figure 2c) in cranberry juice fruit was inappreciable and approximately 1.0 for sugar and above 0.9 for organic acids, with the exception being the G5 membrane. Perhaps the high rejection concealed the effect of feed flow rate. 

Figure 2 also shows the effect of feed flow rate on the benzoic acid separation from cranberry juice. From data in Figure 2b–d, it is apparent that rejection of benzoic acid was lower than for sugars and other organic acids. The reasonable explanation for this is that the molecular weight of benzoic acid is lower than the others. Thus, benzoic acid goes through membrane more easily. Simultaneously, results also indicate that cross flow velocity slightly affected the rejection of benzoic acid. Maybe this phenomenon is related to the contribution of convection and selective layers made from reversible adsorption on the surface of membranes into benzoic acid separation. At a feed flow rate of 3 to 4 L/min, the rejection of benzoic acid in UF-treated cranberry juice decreased, then it increased. This is explained by the UF treatment, which helps cranberry juice reject large compounds so that the small compounds easily pass through the membrane. However, an increase in feed flow rate might promote fouling phenomenon or concentration polarization, leading to higher resistance at the membrane surface [20]. This prevents compounds from passing through the membrane and, consequently, an increase in rejection.

Data from Figure 2a also show that the permeation of UF cranberry juice was approximately two-fold higher than the one of fresh cranberry juice. The possible explanation for this is due to the absorption of high molecular weight compounds and the cake layer on a membrane’s surface. This causes an increase in permeate resistance. Since the cake layer contributed to the separation, the presence of a reversibly adsorptive layer on the membrane surface also results in the higher rejection of benzoic acid in fresh cranberry juice than in UF-treated juice because the cake layer contributed to the separation. The reversibly adsorptive layer is constituted by interaction between high-molecular-weight compounds, such as anthocyanin and membrane material, which is attributed to molecular interactions, such as hydrophobic interactions and hydrogen bonds [21].

### 3.2. Influence of Temperature

The effect of temperature on the performance of cranberry juice nanofiltration is shown in Figure 3. It is apparent that the permeate flux increased and rejection of benzoic acid decreased with the increase in operating temperature (Figure 3a,c). Rejection of organic acid also slightly reduced. The same effect of temperature on solutes was also reported in some of the literature [22,23,24,25]. This can be explained by the expansion of the active layer structure because the higher temperatures increase the mobility of polymer chains, causing the membrane’s porosity and pore size to increase. In addition, higher temperature tends to a more even distribution of organics between the solution and membrane phases, which means less selective partitioning, and as a result, lower rejection. Rejection of sugar did not change because it was concealed by high rejection (approximately 1.0). Although benzoic acid rejection at 40 °C was lower than that at 25 °C, the reduction was not large (Figure 3b). Moreover, nanofiltration at 40 °C spends more energy than at 25 °C. Thus, ambient temperature is suitable for separating benzoic acid by nanofiltration from cranberry juice.

### 3.3. Influence of pH

The influence of pH on the nanofiltration performance to separate benzoic acid from cranberry juice is shown in Figure 4. The permeate flux was slightly affected by pH value. The change in permeate flux relates to the change in pore size and electroviscosity. The change in pH leads to the change in charge on pore walls, which takes account of changes in electrostatic interaction, not only between charged groups in membrane materials but also between these groups and water. Consequently, these interactions lead to the increase in charge on the pore wall, pore size (because of swelling) [26] and electroviscosity [27]. While an increase in pore size increases permeate flux, the increase in electroviscosity makes permeate flux decrease. If the increase in electroviscosity is predominant, permeate flux decreases. On the contrary, if the increase in pore size is predominant, permeate flux increases.

The influences of pH on the rejection of sugars are shown in Figure 4b. From pH 2.5 to 4.5, the separation of sugars is approximately 1. The high rejection of sugar concealed the effect of pH in this range. The rejection tended to decrease slightly when pH increased from 4.5 to 5.5. This can be explained by the increase in pore size because of charge on the pore wall increasing [28,29].

From the data in Figure 4c, the rejection of organic acids tends to increase with an increase in pH, especially in the case of G5 membrane. This phenomenon relates to the repulsion between charged groups in the membrane and solutes. With the increase in pH value, the charge of membranes becomes more negative and organic acid also dissociates more. At pH 5.5, the dissociation of malic acid, citric acid and quinic acid is approximately 100%. Thus, it is difficult for these acids to move through membranes because of the repulsive force between negatively charged groups (on the membrane surface and pore wall) and negatively charged solutes (the dissociated organic acids). Consequently, they were still retained in the retentate side.

With regard to the separation of benzoic acid, from pH 2.5 to 3.5 the rejection decreased, then it increased with an increase in pH. There are some attributes contributing to the change in benzoic acid rejection by nanofiltration membranes under the effect of pH (Figure 4d). Firstly, the pH of cranberry juice was adjusted by NaOH. Thus, the content of sodium ion in the juice increased with the increase in pH. The presence of sodium ion can cause the swelling in membrane, which affects mechanical properties of the membrane simultaneously, altering their ability to recover adsorbed substances [14,30,31]. Thus, solutes can move through membrane more easily and cause a decrease in the rejection. Moreover, the augmentation of sodium content in the feed leads to an increase in sodium content in the permeate side and reinforces movement of the dissociated benzoic acid through the membrane (Donnan effect) [32].

Secondly, this could be due to the effect of pH on the dissociation. The dissociation of organic acid increases with an increase in pH value (Figure 4c). At pH 2.5, only 2% of benzoic acid is dissociated. However, at pH 5.5, 95% of benzoic acid is dissociated. The investigated membranes are made from polyamide or a mixture of amide and the others. Polyamide is amphoteric and its charge depends on pH. The charge is positive with a pH lower than the isoelectric point (pI) and negative with a pH higher than pI. The pI of polyamide membranes often ranges from 4.0 to 6.0 [33,34]. If charge of membrane is positive, then this reinforces dissociated acids to move through the membrane. Conversely, the negative charge in the membrane repulses dissociated acids and increases the rejection. Therefore, from pH 2.5 to 3.5, the charge of membrane was positive and reinforced dissociated benzoic acid to move through the membrane and make rejection decrease. However, with pH rangng from 4.5 to 5.5, membrane charge may be negative, consequently, the repulsive force between membrane and dissociated benzoic acid increased making rejection increase. 

Finally, as stated above, changes in pH can lead to a change in pore size [35]. The intensity of influence depends on the intrinsic membrane, such as its material or structure. These changes in pore size have obvious effects on rejection. 

From pH 2.5 to 3.5, the factors which reinforced the movement of benzoic acid through membranes might be predominant. Therefore, the rejection of benzoic acid decreased. However, with the higher pH, the factors which hinder the transport of benzoic acid through nanofiltration membranes were predominant, especially the repulsive force between membrane and solutes. Consequently, rejection increased with an increase in pH value.

The results also showed that DK and NF99 membranes were the most suitable to separate benzoic acid from cranberry juice because of the high performance of nanofiltration.There were larger differences between rejections of benzoic acid and the others in cranberry juice, and high permeate flux. Besides, the pretreatment by ultrafiltration can improve permeate flux in nanofiltration.

The investigation into the influence of permeate flux on the rejection of benzoic acid from cranberry juice was carried out with NF99 and Desal DK membranes, and the results are shown in Figure 5c,d. The results showed that benzoic acid rejection increased with increases in permeate flux and tended to reach a critical rejection with both NF99 and Desal DK membrane at investigated pH values. As stated above, the transport of solutes through nanofiltration membranes is conducted by three mechanisms: convection, diffusion and electromigration. The proportion of their contribution depends on the attributes of the membrane (for example, material, structure, electrical property), solutes and operative conditions. Based on the extended Nernst–Planck equation and experimental data which investigated model solutions, many authors showed that rejection increased with increases in permeate flux and reached a critical value, so-called reflection rejection, when permeate flux moved towards the infinite, because of the domination of convection [29,33,34,36].

### 3.4. Influence of Permeate Flux

Data in Figure 5c,d also showed that, at low permeate flux and high pH, the rejection of benzoic became negative for the NF99 and DK membranes. This result indicates that, at low permeate flux, diffusion and electromigration considerably contributed to the transport of benzoic acid through investigated nanofiltration membranes. This result accords with those of Szymczyk et al., 2003, obtained by solving theoretical equations [37]. In the case of our study, this can be explained by the influence of sodium ion on the separation of benzoic acid by nanofiltration membranes. When pH was adjusted by NaOH, the sodium content in feed increased with an increase in pH. Consequently, the content of sodium ion in the permeate side increased, and dissociated benzoic acid has to move more to permeate the side to neutralize electricity. Therefore, the content of benzoic acid in the permeate side could become greater than in the retentate side and rejection was negative. However, more research needs to be undertaken to more clearly investigate the transport of benzoic acid through nanofiltration membranes.

Rejections of sugar and organic acid under the effects of pH were also observed (Figure 6). In both NF99 and Desal DK membranes, rejection of sugars and organic acids was over 0.9 and increased with increases in operation pressure.

## 4. Conclusions

The influence of technical parameters on the separation of benzoic acid from cranberry juice by cross-flow nanofiltration with DSS “Lab Module Type 20” plate and frame system was investigated. The pretreatment by UF made permeate flux two-fold higher and separation lower. In the range from 2 L/min to 7.5 L/min, the effect of feed flow rate on performance of separation was slight. Higher temperatures led to higher permeate and lower rejection of benzoic acid, and the suitable temperature for separation of benzoic acid from cranberry juice was ambient. pH strongly affected the performance of nanofiltration, especially benzoic acid rejection, and the lowest rejection of benzoic acid was observed at pH 4.5. When compared to other membranes, the UTC 60 has a lower efficiency in separating benzoic acid from cranberry juice. As the rejection of reducing sugar and organic acid is not significantly different between membranes, rejection of benzoic acid of UTC60 is higher. As a result, the recovery efficiency and purity of benzoic acid are lower. With NF99 and DK membranes, in suitable conditions, the rejection of benzoic acid can be negative; this indicated that benzoic acid has high permeability and the permeate flux is low, while a reduction in sugar and organic acid is retained in the retentate flow. The results showed that cross-flow nanofiltration with NF99 and Desal DK membranes can be applied for the effective separation of benzoic acid from cranberry juice.

## Figures and Tables

**Figure 1 membranes-11-00329-f001:**
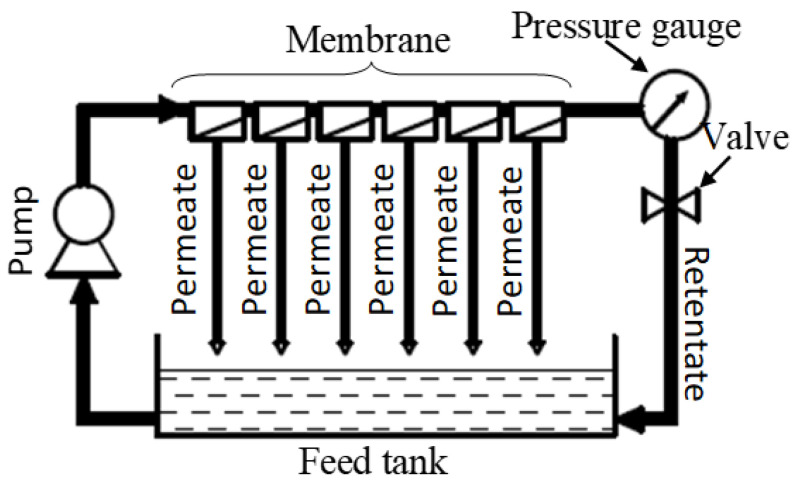
Schema of “Lab Module Type 20” plate and frame system.

**Figure 2 membranes-11-00329-f002:**
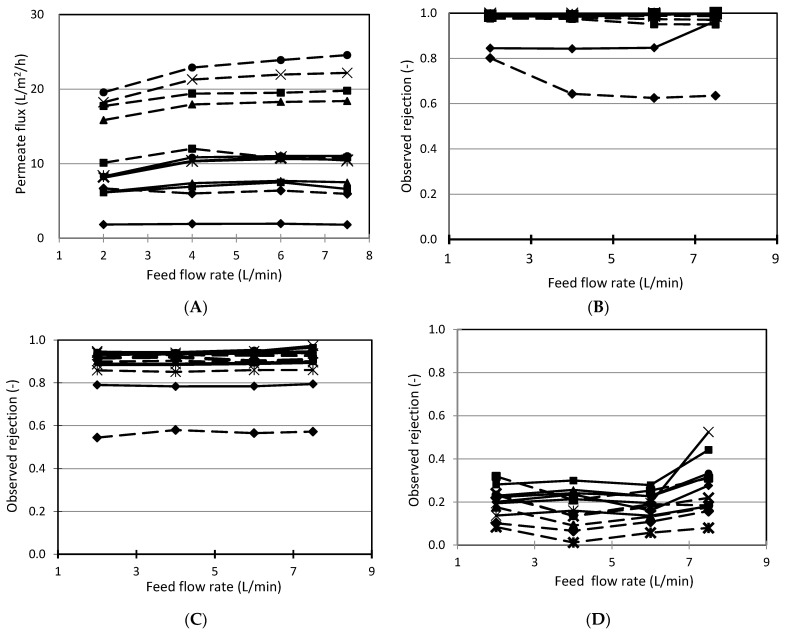
Effect of feed flow rate on nanofiltration of cranberry juice. Solid line: fresh cranberry juice, dashed line: UF cranberry juice. (**A**): permeate flux, (**B**): sugar rejection, (**C**): organic acid rejection, (**D**): benzoic acid rejection. The conditions of nanofiltration: operating pressure: 3 MPa, operating temperature: 25 °C, pH: 2.5. ▲: NTR7250, ■: UTC 60, ♦: G5, ●: DRA 4510, ✕: NF99, 
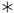
: Desal DK.

**Figure 3 membranes-11-00329-f003:**
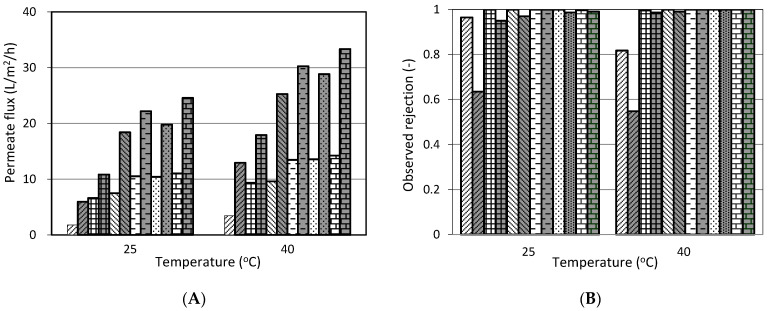
Effect of temperature on nanofiltration of cranberry juice. White background: fresh cranberry juice, grey background: UF cranberry juice. (**A**): permeate flux, (**B**): sugar rejection, (**C**): organic acid rejection, (**D**): benzoic acid rejection. The conditions of nanofiltration: operating pressure: 3 MPa, feed flow rate: 7.5 L/min, pH: 2.5 (na: non-available data).

**Figure 4 membranes-11-00329-f004:**
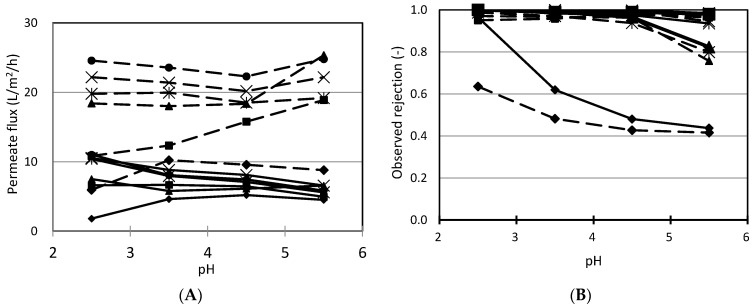
Effect of pH on nanofiltration of cranberry juice. Solid line: fresh cranberry juice, dashed line: UF cranberry juice. (**A**): permeate flux, (**B**): sugar rejection, (**C**): organic acid rejection, (**D**): benzoic acid rejection. The conditions of nanofiltration: operating pressure: 3 MPa, operating temperature: 25 °C, feed flow rate: 7.5 L/min. ▲: NTR7250, ■: UTC 60, ♦: G5, ●: DRA 4510, ✕: NF99, 
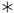
: Desal DK.

**Figure 5 membranes-11-00329-f005:**
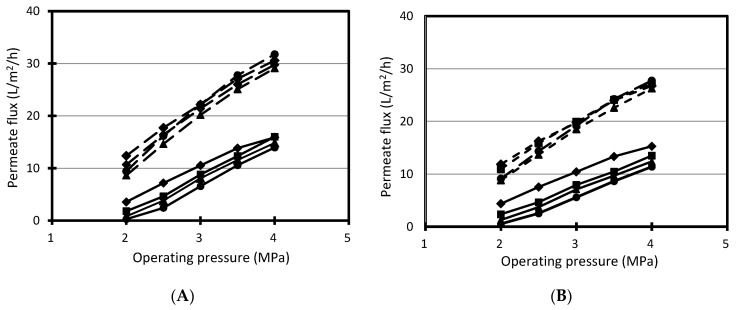
Observed rejection against permeate flux during nanofiltration of cranberry juice. (**A**): permeate flux vs. operating pressure with NF99 membrane, (**B**): permeate flux vs. operating pressure with Desal DK membrane, (**C**): rejection of benzoic acid vs. permeate flux with NF99 membrane, (**D**): rejection of benzoic acid vs. permeate flux with Desal DK membrane. ♦: pH 2.5, ■: pH 3.5, ▲: pH 4.5, ●: pH 5.5. Solid line: fresh cranberry juice, dashed line: UF cranberry juice. The conditions of nanofiltration: operating pressure: 2–4 MPa, operating temperature: 25 °C, feed flow rate: 7.5 L/min.

**Figure 6 membranes-11-00329-f006:**
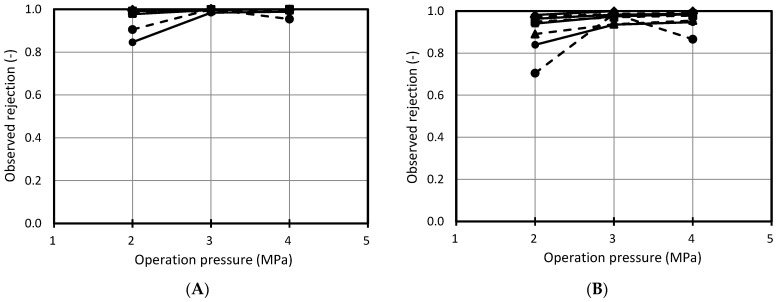
Observed rejection sugars and organic acids against operation pressure. Solid line: fresh cranberry juice, dashed line: UF cranberry juice. (**A**): sugar rejection by NF99, (**B**): sugar rejection by Desal DK, (**C**): organic acid rejection by NF99, (**D**): organic acid rejection by Desal DK. Conditions of nanofiltration: operating pressure: 3 MPa, operating temperature: 25 °C, feed flow rate: 7.5 L/min. ♦: pH 2.5, ■: pH 3.5, ▲: pH 4.5, ●: pH 5.5.

**Table 1 membranes-11-00329-t001:** Characteristics of membranes.

Type	Manufacturer	NaCl Rejection (%)	pH Rank	IEP	Temperature Rank	Permeability (L/m^2^/h/bar)	Material
G5 (GE)	GE	1000 *	2–11	-	<50 °C	0.91	Composite-Polyamide
UTC 60	Toray	55	-	3.2 [16]	-	3.9	Polyamide
NTR 7250	Nitto Denko	60	2–8	-	<60 °C	-	Polyvinyl alcohol
NF99	Alfa-Laval	55	2–10	4.1–4.4 [16]	-	7.00	Composite-Polyamide
Desal-DK	GE	50	2–11	4.7 [17]	<50 °C	2.67	Polyamide
DRA 4510	Daicen	45	2–11	-	-	3.39	Composite-Polyamide

***** based on the molecular weight cut off (Da).

## Data Availability

Data available on request from the authors.

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
