# Peer review of "Influences of Technological Parameters on Cross-Flow Nanofiltration of Cranberry Juice"

_membranes, 2021, doi:10.3390/membranes11050329_

Round 1
Reviewer 1 Report
The manuscript contributes to the optimization of the retention of benzoic acid nanofiltration of cranberry juice, in particular the localization of the optimal pH range of the filtration. The topic is also very interesting economically and technically.
Six commercial nf membranes were examined. These membranes shown in Table 1 are difficult for readers to identify. Manufacturer and name may not be up to date anymore?
More information on cut-off (MWCO) and isoelectric point (membrane charge- or zeta-potential vs. pH) of each membrane would be helpful.
For the description of the charge-dependent exclusion mechanism
these parameters are necessary even if polymeric NF-membranes have similar isoelectric points and are negatively charged between pH 4 and pH 6.
pH-dependent dissociation of benzoic acid (pka=4.21) could also be better described or represented with equilibrium reactions. At pH=4.21 benzoic acid is uncarged.
The charge strength or zeta potential as a function of pH are accessible and published parameters. At least for some well-known NF- membranes in ideal measuring conditions. Influencing parameters are: Membran material (Composite?), Adsorption of Ions or Molecules, Ion streng or conductivity.
These parameters make the real nf very complex. The charge of the membrane and even of the moleculs/acids can change due to the change in pH and possibly due to fouling or adsorption of ions or molecules. The addition of NaOH changes the pH and so does the charge of both the membrane and benzoicacid. The overall ionic strength also increases and, through the Donnan effect, supports the preferred permeation of benzene acid. The addition of Na+- ions also reduces the conductivity, as the concentration of the very mobile H+ ions (very high molar conductivity) decreases.
The measurement of the conductivity would also have been interesting.
If the total conductivity is not too high, this could significantly increase the absolute value of the membrane charge (shielding of the charge is not so strong). Not changing the polarity (Na+ indifferent)! But other specifically adsorbing ions (Ca2+,or SO42- ) or charged molecules (by adsorption or fouling) can also change the membrane polarity and value of charge too.
Not widespread in the industrial NF-process, but there is a possibility for the future investigation to understand the complex charge-determined nanofiltration is monitor the charge online during the process by streaming potential measurements.
The Investigation is a nice example of how these complex interrelationships make economic separation of benzoic acid possible at a pH value of around 4.5. and how important is the pH optimization of the process.
In addition to the rejection, specifying a separation factor or selectivity would also be beneficial.
Reviewer 2 Report
Comments:
In this manuscript, the authors have investigated the effect of operative conditions on separation of benzoic acid (a natural preservative) from Cranberry juice. Authors studied six different commercially available nanofiltration membranes for above mentioned application.
The results presented are careful evaluation of operating conditions for separation of benzoic acid. In my opinion, the manuscript can be published in Membranes after below mentions queries are satisfied satisfactorily. I suggest major revision for this manuscript.
- The definition of nanofiltration membrane (NF) (line 49) is generic. The manuscript is written for expert readers and more specific definition is expected. Please define the NF membrane based on specific solute size or molecular cutoff range, operating feed pressure range (can be found out in literature, example: https://doi.org/10.1021/cr500006j, Figure 4) and relate it to benzoic acid separation application studied in this paper, rather than defining it as a membrane with MWCO between UF and RO membranes.
- Check line 59 in introduction. Size exclusion or MWCO does not depend on thickness of active layer of the membrane BUT flux is dependent on thickness. Size exclusion is only dependent on pores and pore size. Authors should provide citation for this statement.
- In Table 1, how G5 (GE) has a NaCl rejection of 1000 %? Does author mean to write 100 %?
- In Figure 2C what organic acids are expected? Do authors have structure/ mol. wt. information of these acids?
- In Figure 2D, why decrease in rejection of benzoic acid (UF treated cranberry juice; dashed line) is observed for feed flow rate around 4 L/min compared to other flow rates (higher as well as lower flow rates)? This is observed for all membranes.
- Please provide reference for statement in line number 173-176.
- Although authors have relied heavily on supporting the observation of decrease in rejection at high temperatures, it’s worth checking the DSC of polyamide (can be found in literature) and verify if chain mobility can occur at the temperatures at which the test were carried out.
- The NFT50 membrane has flux around 7.00 Lm-1h-1bar-1 whereas DRA 4510 has almost half the flux of NFT50 (as per the values in Table 1), yet the permeate flux in Figure 2A shows NFT50 having less flux compared to DRA4510. This is also observed for high temperature (25 as well as 40 deg) experiments (Figure 3A), as per author’s understanding, what is the explanation for this? As per authors if the high temperature is making the polymer chain mobile and increasing the pore size then NFT50 should show highest flux.
- The temperature dependent rejection of benzoic acid (Figure 3D) NFT 50 shows highest rejection even though the MWCO for G5 is 100 %. It is interesting and authors should try to explain this.
- In the influence of pH section, it will be worth mentioning the charges on membranes at which the test are done. The charges can be either measured using streaming potential instrument for found out in literature.
- What is “electroviscosity”? Do author want to write “Visco-electric effect”?
- What are sugars been separated from? (line number 208). It should be rejection not separation?
- “The presence of sodium ion can cause the swelling in membrane, which 229 makes membrane more porous and pore size larger [22–24]” (line 229-230). Swelling does not make the membrane more porous, it does increase the pore size only. Check and change the sentence.
- Authors mentioned that polyamide membranes are amphoteric and charge depends on pH. It is true but not for all cases. With data not mention for charges on membrane (zeta potential/streaming potential curve), authors should correlate this with their membranes where chemical structures for the membranes are not specified.
- Please cite the figures when discussing the data. For example from line 234 onwards, I had to repeatedly go back to the figures and search and correlate with the data discussed. This practice is found through-out the manuscript. Please rectify.
- “Finally, as stated above, change in pH can lead to change of pore size” PLEASE PROVIDE REFERENCE. I agree that charge may change on pore walls w.r.t pH but polyamide NF membranes do not change pore size with pH. These membranes are not pH responsive w.r.t. to pore size.
Other recommendations:
- Grammatical errors are found everywhere in the manuscript and I have mentioned some of the line where changes needs to be done. Recommend thorough editing of the manuscript, get it checked by native English speaker.
- Line 76 in intro “lead to fouling” replace with “leading to fouling” or rephrase the sentence to fit well, grammatically.
- For all the figures, the numbering (“A”, “B”….) is given at the bottom right of the figure. I recommend to follow a standard publishing practice and put the numbering on top right. It will be easy to read.
- In the results section, starting from line 143 to 176, authors should mention Figure 2A, 2B and not just Figure 2. It’s difficult to go to figure and locate the correct one each time.
- “It makes the resistant to permeate increase” rephrase to “increasing the permeate resistance”.
- Please check sentence in line number 173, does not sound correct grammatically.
- Rephrase line 203 and 204 “Whereas, increase in 203 pore size makes permeate flux increase, the increase in electroviscosity makes permeate 204 flux decrease”, grammatically does not reads corrects.
- Line 212 “the rejection of organic acids tended to increase” tended should be “tends” or rephrase.
- “And, they were still retained in 219 retentate side” rephrase Please don’t start the sentence with “And”. Also replace “will be retained in retentate side” with will be in rejected.
- Line 230 “Thus, solutes can move 230 through membrane more easily and make rejection decrease” what is “make rejection decrease”? Do authors mean to say “decrease the rejection”?
- “the rejection is increase” line 242. Checked grammar.
Round 2
Reviewer 2 Report
Accept the manuscript in current format.
Author Response
We are grateful for your consideration. Thank you for your recommendations.